# Special Issue about Head and Neck Cancers: HPV Positive Cancers

**DOI:** 10.3390/ijms21093388

**Published:** 2020-05-11

**Authors:** Panagiota Economopoulou, Ioannis Kotsantis, Amanda Psyrri

**Affiliations:** Section of Medical Oncology, Second Department of Internal Medicine, National and Kapodistrian University of Athens, Attikon University Hospital, 12462 Athens, Greece; panagiota_oiko@hotmail.com (P.E.); ikotsantis@gmail.com (I.K.)

**Keywords:** head and neck cancer, HPV, oropharyngeal cancer, de-escalation

## Abstract

The oropharynx has become the leading primary site for Human Papilloma Virus (HPV)-associated head and neck cancer. HPV positive oropharyngeal squamous cell carcinoma (HPV+ OSCC) has emerged as an epidemic not easily recognized by many physicians, resulting in delays in diagnosis and management. HPV+ OSCC traditionally refers to younger, healthier patients with high economic status and high-risk sexual behavior and is related to improved prognosis. De-intensification strategies are being evaluated in ongoing clinical trials and if validated, might help spare severe morbidity associated with current cisplatin-based chemoradiotherapy, which is the standard of care for all patients with locally advanced head and neck cancer. On the other hand, whether HPV status represents an important prognostic factor for non-oropharyngeal sites remains to be elucidated.

## 1. Introduction

High-risk Human Papillomaviruses (HPVs) are associated with the development of premalignant lesions and certain cancers due to their high oncogenic potential. Indeed, HPV is involved in head and neck cancer carcinogenesis and represents an important contributing factor particularly in cancer of the oropharynx. In developed countries, HPV currently accounts for the majority of patients with primary oropharyngeal squamous cell carcinoma (OSCC), taking precedence over traditional etiological factors, such as smoking and alcohol [1]. Most importantly, HPV positive OSCC (HPV+ OSCC) comprises a distinct disease entity, with a different clinical and biological behavior [2]. Although current data are insufficient to suggest changes in management of HPV+ OSCC, evaluation of HPV status has been incorporated in treatment guidelines worldwide due to a major impact in prognosis which is also reflected on the new Tumor Node Metastasis (TNM) staging for HPV+ disease [3]. Whether HPV status will define the type of treatment approach in the near future is being evaluated in numerous ongoing clinical trials and remains to be determined. In this review, we will focus on the prognostic and predictive role of HPV in OSCC and illustrate the clinical characteristics and management of HPV+ OSCC. In addition, we will discuss the utility of assessing HPV status as a clinically relevant biomarker in HPV-associated non-oropharyngeal head and neck squamous cell carcinoma (HNSCC).

## 2. HPV Life Cycle 

HPV life cycle is highly dependent on the cellular procedures of the host cell. Initially, HPV virus infects actively dividing basal keratinocytes and is subsequently subjected to viral genome amplification [4]. Viral genome is maintained as an extrachromosomal element which is called episome [5,6]. High-risk HPVs, such as HPV16 and HPV18 that contribute to HPV-induced carcinogenesis, contain two important viral oncoproteins E6 and E7. Like other host-dependent viruses, HPV uses the host cell replication machinery to initiate viral DNA replication, which is consequently promoted by E6 and E7 [7]. Subsequently, HPV exploits the host replication machinery to amplify its viral genome; oncoproteins E6 and E7 are essential for protection of the viral genome and re-entry of the host cell to S-phase [8]. In addition, both oncoproteins E7 and E6 contribute to the formation of a cellular microenvironment that favors viral genome replication. E6 protein provokes the degradation of the p53 tumor suppressor protein [9]. On the other hand, E7 oncoprotein mediates the degradation of retinoblastoma (Rb) tumor suppressor protein that negative regulates p16, a protein encoded by the *CDKN2A* gene (cyclin dependent kinase inhibitor 2A) [10]. P16 normally blocks activation of Rb and stops proliferation of damaged cells. Degradation of Rb causes overexpression of p16, which occurs distinctively in HPV+ OSCC. Consequently, HPV+ OSCC is characterized by p53 degradation, Rb down-regulation and p16 up-regulation. [11,12]. On the contrary, smoking-related OSCC is defined by mutations in *p53*, Rb up-regulation and p16 down-regulation [13]. 

## 3. Molecular Aspects of HPV+ OSCC

Interestingly, molecular data from recent genomic studies demonstrate that HPV+ and HPV negative (HPV-HNSCC) have distinct genomic profiles. The Cancer Genome Atlas (TCGA) reported the first results of sequencing data from HNSCC biopsies in 2015 [14]. It is widely known that tobacco-associated HPV-HNSCC has a high mutational load and is characterized by multiple molecular alterations, such as deleterious mutations of or loss of *p53* (84%), mutation or loss of *CDKN2A* (58%), amplification of *CCND1* (31%), which is an oncogene implicated in cell cycle regulation, amplification of *MYC* (14%) and overactivation of PI3K kinase pathway (30%) [15,16]. On the contrary, although the TCGA data included a small number of HPV+ cases, it becomes increasingly clear that HPV+ cancers have a far less complex genomic profile. More specifically, they rarely harbor *TP53* mutations (3%) or *CDKN2A* alterations (0%). The aforementioned E6 and E7 viral proteins functionally inactivate p53 and pRb in HPV+ HNSCC [10]. In addition, compared to tobacco-related HNSCC, HPV+ cancers display more commonly gains in chromosome 3q and losses of chromosomes 11q, 13q, 14q, 16p and 16q [17]. On the contrary, HPV− cancers exhibit more frequently amplifications of chromosome 7p and deletions of chromosome 9p that contain the *EGFR* and *CDKN2A* genes. Moreover, mutations or alterations in the *PI3KCA* gene are commonly observed in HPV+ OSCC. On the other hand, HPV+ tumors are characterized by loss of *TRAF3* (TNF Receptor-Associated Factor 3) (22%) and amplification of *E2F1* (19%) [13,14].

Of note, there has been a trend to categorize HPV+ OSCCs in clinical and prognostication subgroups based on their gene expression clustering. The classical or keratinocyte differentiation (HPV KRT) subtype includes cancers with primarily keratinocyte differentiation, higher proportion of spliced *E6* oncogene, overactivation of *PI3KCA* and worse outcome, whereas the inflamed mesenchymal (HPV IMU) subtype is mainly correlated with profound immunogenicity and mesenchymal cell differentiation. However, this classification has not yet been incorporated in clinical practice [18,19]. In addition, a second stratification has been created based on p16 expression and detection of HPV DNA. Thus, patients included in Class I subgroup have low p16 levels and absent HPV DNA, those included in Class II subgroup are characterized by low p16 levels but presence of HPV DNA and those included in Class III subgroup demonstrate high p16 expression and presence of HPV DNA. Obviously, patients that belong to Class III subgroup have better prognosis, as high p16 expression is an independent prognostic factor [12]. However, the clinical relevance of these stratification groups needs to be evaluated in further studies.

Important data also suggest a role of HPV integration. As described in detail earlier, initially in the HPV life cycle, HPV DNA is found in episomes. Carcinogenesis involves the integration of HPV viral DNA into the host cell genome, which leads to high expression of viral oncoproteins E6 and E7 [20]. However, it is unclear how HPV integration affects the host genome. Initial studies in HNSCC cell lines have demonstrated a correlation between HPV integrants and large genomic alterations of the host [21]. In addition, Parfenov et al. performed whole genome sequencing, transcriptome, and DNA methylation analyses of 35 HPV+ HNSCCs and found that HPV is identified as both integrated and episomal. Importantly, it was shown that HPV integration impacts host genes by promoting deregulation of tumor suppressor genes, DNA amplification and generation of modified transcripts [22].

Genome-wide association analysis has demonstrated that genomic factors contribute to susceptibility to HPV+ OSCC. More specifically, the Class II haplotype DRB1*1301-DQA1*0103-DQB1*0603 has been shown to have a protective effect over HPV+ OSCC as compared to HPV− OSCC [23]. These genomic differences might interpret dissimilarities in response in treatment and outcome between patients.

Most importantly, HPV genomics that refers to sequencing of the HPV genome has emerged as an interesting field of research and has provided robust data regarding HPV16 genetic variation and cervical carcinogenesis, molecular characterization and characteristics of cervical cancer [24,25]. Future studies of HPV genomics and OSCC might give insight into molecular aspects and epidemiology of OSCC.

## 4. HPV Detection Methods

Determination of HPV status in OSCC is fundamental both as a contributor of diagnosis and a prognostication factor. HPV status has been established as an important and versatile biomarker and routine HPV testing is universally recommended by the College of American Pathologists (CAP), the American Joint Committee on Cancer (AJCC)/Union for International Cancer Control (UICC) and the National Comprehensive Cancer Network (NCCN) guidelines. CAP released formal guidelines in 2018 [26]. Some crucial guidelines include: (1) Testing of all OSCC patients for p16; p16 positivity is equivalent to strong nuclear and cytoplasmic staining in 70% or more of malignant cells, (2) No need for testing for non-oropharyngeal squamous cell carcinomas or for non-squamous OSCC cases and (3) No need for testing low-risk HPV.

HPV status is currently an obligatory stratification factor in clinical trials of OSCC and it is used to determine eligibility for trials evaluating treatment de-escalation. Surprisingly, a substantial proportion of OSCC cases in the US remain untested, mainly due to lack of robust predictive significance and cost-effectiveness. In a recently published study that used National Cancer Database, it was revealed that 12% of OSCCs in the US were not tested between 2013 and 2015, mainly due to insurance issues (patients with private insurance were more likely to be tested) and diagnosis in low-volume hospitals, as large or academic centers were more likely to recommend testing. In addition, testing was performed less frequently in women and older people [27].

Several methods are used to detect the presence of HPV including p16 immunohistochemistry (IHC), HPV DNA in situ hybridization (ISH), *E6*/*E7* HPV RNA-ISH, HPV DNA polymerase chain reaction (PCR) and *E6*/*E7* HPV RT-PCR [28]. P16 IHC identifies p16 overexpression, which is a surrogate marker for HPV status. Despite important advantages, such as wide availability, high sensitivity, and low cost, p16 ICH is rather regarded as a prognostic tool and it cannot be used as a sole test for evaluation of HPV status [29]. Several disadvantages of p16 testing include low specificity in surgical specimens, increased p16 expression by non-viral associated mechanisms which leads to false positive results, differences in interpretation between pathologists and inconsistency between 16 status and actual HPV status [30].

Regarding the remaining HPV-specific testing modalities, each of them has several advantages and disadvantages. It is currently clear that DNA detection alone is not adequate for evaluation of HPV status, mainly due to low sensitivity [26]. Most importantly, detection of *E6*/*E7* mRNA via PCR in tissue is deemed the gold standard, given the high sensitivity and feasibility of this method; nevertheless, its sensitivity is significantly decreased once the sample is of poor quality. RNA-ISH is currently being used more frequently, particularly in combination with p16 [26]. Obviously, the combination of two strategies for determination of HPV status is a broadly accepted methodology. Most commonly, this approach involves p16 IHC as initial strategy [31].

Subsequently, HPV detection should be confirmed in p16+ cases using RNA PCR or HPV DNA detection strategies. In a study by Schache et al., the authors used several detection strategy combinations to evaluate HPV16 status in fixed and fresh frozen tissue derived from 108 OSCC cases. It was found that the combination of p16 IHC and DNA quantitative PCR (qPCR) yielded the highest sensitivity (97%) and specificity (94%) compared to the gold standard RNA qPCR [32]. Unfortunately, proper evaluation of HPV status is hampered by lack of applicability and cost-effectiveness of multimodality approach. Nevertheless, evaluation of HPV status is extremely important in OSCC patients, mainly due to determination of prognosis and patient selection in treatment de-intensification trials. Additionally, it has been postulated that high-risk HPV+ cases other than HPV16+ have less favorable prognosis [33].

## 5. HPV+ OSCC

### 5.1. Epidemiology and Clinical Characteristics

Despite a steady decline in the incidence of non-oropharyngeal HNSCC during recent decades, we have been witnesses of an overall increase in the incidence of OSCC following the rise of OSCC cases attributed to HPV infection [34]. From 1975 through 2012, a 26.6% increase in incidence of OSCC has been reported worldwide, whereas in the USA the prevalence of HPV+ OSCC has been reported to have increased by 225% [35]. Currently, it has been postulated that approximately 90% of OSCC cases in the USA are HPV-driven. Clearly, the proportion of HPV+ cases is elevated (40%–60%) in western countries, such as North America, Northwestern Europe, South Korea, Japan and Australia and low (13%–24%) in South Europe, China, and India, mainly owing to the high percentage of oral sex and multiple sexual partners observed in western countries. In the future, it is likely that HPV+ OSCCs will account for the majority of head and neck cancers [36,37].

HPV16 accounts for almost 90% of OSCC cases [38]. This subtype is also most commonly encountered in cervical cancer. However, other oncogenic subtypes, such as HPV33, HPV18 and HPV31 have been less commonly implicated in the pathogenesis HPV+ OSCC. Of note, it seems that patients with HPV+ OSCC attributed to less common HPV subtypes have less favorable prognosis [33].

Clinical presentation of HPV+ OSCC differs significantly from tobacco-related HNSCC, which traditionally affects older adults with tobacco or alcohol abuse [39]. On the contrary, HPV+ OSCC is classically seen in younger men with little or no history of alcohol abuse. More specifically, median age at presentation is 57 years old as compared to 64 years old for HPV− OSCC. However, a subset of patients is diagnosed at a later age. In a recently published prospective study, it was revealed that younger patients (<50 years old) have a different sexual behavior that older patients, which includes a higher number of oral sex partners per year and increased sexual intensity [40]. Nevertheless, a direct association between high-risk sexual behavior and HPV+ OSCC has not been established. A recent review that included 20 studies concluded that data is inconsistent. Despite this inconsistency, the number of sexual partners and oral sex probably pose a greater risk of OSCC [41]. Importantly, patients with HPV+ tumors tend to metastasize to cervical lymph nodes, which frequently exhibit necrosis and cystic changes. Interestingly, the extensive involvement of cervical lymph nodes is not predictive of dismal prognosis and these tumors usually exhibit good locoregional control when managed with combined cisplatin-based chemotherapy and radiation [42]. This matter has been confronted in the latest edition of AJCC staging which endorses a separate staging of OSCC based on HPV status.

HPV+ OSCC display a significantly better prognosis compared to tobacco-related HNSCC and the risk of death has been reported to be approximately 60% of that of HPV− OSCC [43]. Improved survival is mainly attributed to better locoregional control, since the incidence of distant metastases is similar regardless of HPV status. Of note, smokers have a worse prognosis than patients with no history of tobacco consumption [43].

### 5.2. Staging 

The latest edition of the AJCC/UICC tumor classification system has been reported in 2018 after assessment of research focused on molecular pathogenesis and prognostic factors of head and neck cancer [3]. The development of a separate staging system for OSCC based on HPV status mirrors the markedly improved different prognosis of HPV+ OSCC. More specifically, clinical characteristics such as number and size of clinically positive lymph nodes and the presence of extracapsular extension represent earlier stages in HPV+ tumors as compared to HPV− cases. For example, bilateral/contralateral or large size lymph nodes (>6 cm) are considered to be stage II and III respectively in HPV+ OSCC as compared to stage IVA and IVB in HPV− OSCC. In addition, distinct clinical and pathological N-definitions have been incorporated in the new classification [44]. The clinical TNM staging is based on the proposal of the International Collaboration on Oropharyngeal cancer Network (ICON-S) group and is originated from a study that included 1907 HPV+ OSCC using training (PMH) validation [45]. The pathologic TNM stage was derived from a study that included 704 surgically treated HPV+ patients [46].

This emerging HPV-specific classification, which represents the urgent need to appropriately portray the nature and prognosis of the new disease, is essential in many aspects. First, it is useful for stratification in de-escalation trials focused on HPV+ disease; second, it is clinically relevant for discussion with patients to properly understand the stage and prognosis of the disease. Last, it is a necessary tool for the future, as separate management recommendations may be applicable in HPV+ and HPV− tumors.

The 8th edition of the clinical and pathological TNM staging is illustrated in Table 1.

### 5.3. Treatment and De-Escalation Strategies

In western countries, the gold standard for patients with localized or locally advanced OSCC is combined cisplatin-based chemotherapy and radiation, excluding patients with early-stage disease who may be qualified for surgical intervention. Chemoradiation has demonstrated impressive results, with a 5-year overall survival of 95% in HPV+ patients [47]. In developing countries where there is a shortage of radiation facilities, the majority of patients are subjected to surgery [48]. However, patients affected with HPV+ disease are traditionally younger with a low burden of co-morbidities and have a significant survival advantage compared to patients with HPV− OSCC. Thus, the “one size fits all” approach might not have been appropriate for OSCC and the question that needed to be addressed is whether morbidity associated with standard chemoradiotherapy could be diminished without jeopardizing effectiveness of treatment. Consequently, subsequent trials in HPV+ OSCC attempted to evaluate the so-called “treatment de-escalation strategies” aiming to reduce morbidity and prevent overtreatment.

The idea of replacing cisplatin, a toxic chemotherapy agent, with cetuximab, a monoclonal antibody against EGFR in locally advanced OSCC has been developed rather early, after the IMCL 9815 trial, that compared cetuximab in combination with radiotherapy with radiotherapy alone has shown good effectiveness in patients with OSCC [49]. In this trial, although the combination of cetuximab and radiation yielded superior survival without major toxicities, the lack of a direct comparison with cisplatin questioned the magnitude of results. Since the Food and Drug Administration (FDA) approval of cetuximab for locally advanced HNSCC, it has been mainly used in patients who were ineligible to receive cisplatin based on specific criteria. In 2019, two large randomized trials addressing this issue have been simultaneously published in the Lancet and results of the two trials have been thoroughly discussed at meetings the whole year [50,51]. The phase III randomized De-Escalate HPV trial randomized 334 patients with HPV+ disease to receive either 3weekly cisplatin in combination or cetuximab at standard doses in combination with Intensity Modulated Radiation Therapy (IMRT) (70 Gy in 35 fractions in 7 weeks) [50]. The primary endpoint was a reduction in grade 3–5 adverse events in the cetuximab arm and secondary endpoints were OS and rate of relapse between treatment arms. The primary endpoint was not met, since no statistically significant difference was demonstrated between the two groups regarding severe acute and late toxicities. In addition, cetuximab was correlated with worse OS, locoregional and distant control [50].

Herein, we discuss several pitfalls of the De-Escalate HPV Trial. First, it included only patients with low-risk “favorable” HPV+ OSCC subgroup. This group selection was arbitrarily based on two parameters: a) the 7th edition of TNM AJCC/UICC staging system (T3–T4N0, and T1N1–T4N3) and b) the stratification by Ang et al. in their analysis of the RTOG 0129 study cohort of HPV+ OSCC patients [43]. Ang et al. stratified patients in three categories according to N stage and history of smoking [43]. Thus, in the De-Escalate HPV trial, “low-risk” patients were not selected based on an HPV-specific staging system (8th edition); Second, there was poor compliance in the cisplatin arm, since only 38.3% of patients completed the full protocol due to hematologic and gastrointestinal toxicity. The corresponding percentage in the cetuximab arm was 79%; thus, safety results may be biased due to differences in compliance; Third, survival data were probably immature, since there were only 26 deaths with 19 cancer-related deaths during follow-up; Last, the protocol of radiotherapy used in the De-Escalate HPV (once daily fractionation) was different than that used in the Bonner trial (either daily fractionation, twice-daily fractionation or concomitant boost radiotherapy) [49,50]. Although the Bonner trial was not statistically powered to report differences between, since groups, survival was mainly improved in patients that received concomitant boost [49]. Therefore, poorer survival observed in the cetuximab arm of the De-Escalate Trial might be attributed to type of radiotherapy used in the trial.

On the other hand, Gillison et al. reported the results of the RTOG 1016 trial, a non-inferiority trial that randomized 849 “all comers” with HPV+ OSCC to receive either cisplatin or cetuximab in combination with IMRT [51]. Although the trial was not restricted to a low-risk population sample, 71% of included patients were low-risk and 29% were intermediate risk based on stratification by Ang [43]. In addition, a small percentage of patients were T4 (12%) or N3 (4%) disease. Primary endpoint was OS. RTOG 1016 was a negative trial, since the non-inferiority criteria was not met after 4.5 years of follow-up (HR = 1·45, one-sided 95% upper CI 1·94; *p* = 0·5056 for non-inferiority; one-sided log-rank *p* = 0·0163) and patients in the cisplatin arm showed superior OS (5-year OS 84.6% vs. 77.9% in the cetuximab arm) [51]. In addition, the rate of grade 3–4 events did not differ between the two arms; however, as expected, several adverse events occurred more commonly in the cisplatin arm, such as myelotoxicity, gastrointestinal toxicity, and nephrotoxicity. Compared to the De-Escalate trial, in the RTOG 1016 trial, (a) patients received two cycles of cisplatin instead of three; (b) a different type of radiation was used (70 Gy in 35 fractions over 6 weeks at six fractions per week); (c) OS was the primary endpoint (vs. the proportion of gr 3–5 adverse events in the De-Escalate trial) and (d) analysis was done after longer follow-up (5 years vs. 26 months in the De-Escalate trial) [50,51].

These two randomized trials that yielded rather disappointing results have clearly shown that it is not the proper time, at least not yet, for de-intensification strategies for any HPV+ patient outside of clinical trials. However, this should not be the end of treatment de-escalation in locally advanced HPV+ OSCC. The lack of predictive biomarkers of response to cetuximab hampers proper design of clinical trials. However, other de-intensification strategies have shown good results in phase II trials. Chera et al. conducted two trials evaluating reduced radiation (60 Gy/30 fraction at high-risk areas and 54 Gy at subclinical areas of the neck) in patients with low-risk HPV+ OSCC (N0–N2, T0–T3 based on the AJCC TNM 7th edition) [52,53]. In the first trial, RT was given in combination with weekly cisplatin 30 mg/m^2^ and LN dissection was required after chemoradiation because pathologic complete response (pCR) was the primary endpoint, whereas in the second trial RT was administered in combination with either weekly cisplatin or weekly cetuximab and primary endpoint was 2-year Progression Free Survival (PFS) [52,53]. Both these trials showed 3-year distant free survival (DFS) and Overall Survival (OS)ranging from 91% to 100% and 95%, respectively; approximately 35% of patients had grade 3+ mucositis and dysphagia. In addition, four phase II trials assessed the efficacy of induction chemotherapy (IC) followed by reduced dose RT in either complete responders [54], complete and partial responders [55,56], or ≥50% responders [57] to IC and included both low and high-risk HPV+ categories. These trials demonstrated a 2-year PFS and OS ranging from 80% to 98% with grade 3+ toxicities ranging from 9% to 30%. Randomized phase III trials are warranted for validation of results. Most importantly, there is an urgent need for the discovery of predictive biomarkers that will guide patient selection.

## 6. HPV+ Non-Oropharyngeal Cancer

It is widely known that HPV and p16 are prognostic biomarkers for HPV+ OSCC. However, several studies have pointed out the prevalence of HPV infection in non-oropharyngeal areas, such as the oral cavity, larynx, and oropharynx. More specifically, 3%–5% of oral cavity and laryngeal cancers are HPV+ [58]. The question that arises is whether p16 is a surrogate marker for a transcriptionally active HPV in non-OSCC HPV+ HNSCC. In addition, it is unknown whether p16 and HPV are prognostic factors in non-OSCC HPV+ HNSCC. In a study published in 2011, Harris et al. evaluated the incidence of HPV and p16 positivity in young patients with oral cavity cancer using p16 IHC and HPV ISH and PCR. In this cohort of patients, p16 was expressed in 11 out of 25 patients and p16 expression was significantly associated with relapse free survival (RFS) and OS benefit. However, HPV16 was detected by PCR only in two tumor samples and not detected by ISH in any sample examined [59]. In another retrospective study in a larger cohort of patients with oral cavity cancer (409 patients), HPV was determined using high-risk (HR)-HPV *E6*/*E7* oncogene expression by RT-PCR and p16 was assessed by IHC. It was shown that 3.7 % of patients were HPV16 positive and 2.2% were positive for other HR HPV subtypes. P16 IHC had good sensitivity (79.2%), excellent specificity (93%) and high negative predictive value (98%) but low positive predictive value (41.3%) for HPV detection [60].

Studies suggest that there are differences regarding the role of p16 and HPV positivity in non-OSCC HNSCC. Chung et al. assessed HPV status by ISH and p16 expression by IHC in patients with non-OSCC HNSCC (laryngeal, oral cavity and hypopharyngeal cancer) included in three RTOG studies (0129, 0234, 0522) [61]. It was shown that p16 was not a good surrogate marker for HPV positivity, since it was positive in 14.1%, 24.2% and 19% of cases in the RTOG 0129, 0234, and 0522 respectively compared to 6.5%, 14.6% and 6.9% of cases with HPV ISH positivity. In addition, p16 was found to be a prognostic biomarker for non-OSCC HNSCC; however, it is not so profoundly associated with improved outcome compared to HPV+ OSCC. Of note, p16 negative OSCC and non-OSCC cases have similar OS and PFS. HPV positivity was not shown to be prognostic in non-OSCC cases [61]. In another study that was focused on laryngeal cancer, 324 samples were assessed for p16 status by IHC and HPV status by RNA-ISH. The incidence of p16 positivity was 6.9% and only seven cases were found to be HPV+. Interestingly, neither p16 nor HPV were prognostic [62].

More recently, Fakhry et al. retrospectively tested HNSCC cases diagnosed between 1995 and 2012 for HPV status using DNA and RNA-ISH and p16 status by IHC. In non-OSCC patients, p16 and HPV status were not found to be prognostic [63]. Similarly, De Souza et al. showed no significant prognostic role of HPV/p16 status in 845 non-OSCC cancer patients from Brazil, US and Europe [64]. In contrast, Ko et al. examined a large cohort of patients with non-OSCC HNSCC (19,993) from the National Cancer Database (NCD) and showed a significant survival benefit in HPV+ non-OSCC patients [65]. This survival benefit was confirmed only for hypopharyngeal and locally advanced laryngeal primaries in a retrospective analysis of patients with known HPV status included in NCD who were diagnosed between 2010 and 2013 [66].

Altogether, studies focusing on prognostic impact of HPV status in non-OSCC HNSCC have shown conflicting results. In addition, p16 has been shown to be a poor surrogate biomarker for HPV infection in these cases. Thus, routine HPV testing is not recommended for non-OSCC cases. Prospective studies are required to elucidate the potential role of HPV status in non-OSCC HNSCC.

## 7. Conclusions

HPV+ OSCC currently represents a global epidemic, a widely recognized increasingly prevalent entity. Similar to cervical cancer, the majority of HPV+ cases are attributed to high-risk HPV16 subtype. P16 positivity serves as a surrogate marker for HPV infection and is an independent prognostic factor for those patients. In addition to the differences in risk factors and clinical presentation, HPV+ OSCCs display a substantially improved survival. Thus, HPV status is now routinely used as a stratification factor in clinical trials and is a clinically relevant parameter for patient consultation. This has been depicted in the new edition of AJCC/TNM staging that includes a separate staging manual for HPV+ OSCC. However, randomized trials evaluating de-intensification strategies substituting cisplatin with cetuximab have failed to show non-inferiority in terms of outcome and current guidelines do not recommend a different treatment approach for HPV+ OSCC. HPV status has no confirmed prognostic role for non-OSCC HNSCC. Future studies are necessary to establish the role of de-escalation treatment approaches in HPV+ OSCC. Until then, the cornerstone of treatment is an interdisciplinary team approach with emphasis on quality of life during the treatment and survivorship periods.

## Figures and Tables

**Table 1 ijms-21-03388-t001:** TNM staging for HPV+ oropharyngeal cancer (8th edition). LNs = Lymph Nodes, OSCC = Oropharyngeal Squamous Cell Carcinoma, TNM = Tumor Node Metastasis.

HPV+ OSCC	Clinical Stage			Pathologic Stage		
Category	T	N	M	T	N	M
Stage I	T1, T2	N0: No regional LNs	M0	T1, T2	N0: no regional LNs	M0
		N1: Ipsilateral LNs			N1: 1–4 LNs	
Stage II	T1, T2	N2: bilateral or contralateral LNs	M0	T1, T2	N2: ≥ 5 LNs	M0
	T3	N0: no regional LNs N1: ipsilateral LNs N2: bilateral or contralateral LNs	M0	T3, T4	N0: no regional LNs N1: 1–4 LNs	
Stage III	T4	Any N	M0	T3, T4	N2: ≥ 5 LNs	M0
	Any T	N3: > 6 cm LN(s)	M0			
Stage IV	Any T	Any T	M1	Any T	Any N	M1

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
