# Peer review of "Special Issue about Head and Neck Cancers: HPV Positive Cancers"

_ijms, 2020, doi:10.3390/ijms21093388_

Round 1
Reviewer 1 Report
I don't think that this manuscript adds anything new to the field.
Furthermore, there are editorial errors throughout. The most glaring is stated in line 12 of the abstract (after specifically defining HPV+OPSCC), “HPV-OSCC traditionally refers to younger, healthier patients with high economic status and high-risk sexual behavior and is related to improved prognosis.” I believe that this should be “HPV+OSCC traditionally refers to younger, healthier patients with high economic status and high-risk sexual behavior and is related to improved prognosis.”
This was repeated in the text in line 240, when referring to the De-Escalate trial, “…it included only patients 239 with low-risk “favorable” HPV-OSCC subgroup.” Again, I believe this should read, …it included only patients 239 with low-risk “favorable” HPV+OSCC subgroup.”
The authors use HPV-OSCC throughout the rest of the paper when referring to HPV-negative OSCC. This oversight (particularly in the abstract!!) highlights the lack of attention to detail in this manuscript.
Other errors include:
- Inconsistent use of appropriate formatting of gene and protein names/symbols (italics, capitalization).
- Multiple reference formats (sometimes with Author last name/Journal/Year- these are not associated with the reference list, other times with a number- these are associated with the reference list)
- Et al. is an abbreviation of “et alia” and requires a period after al. This is missing throughout the paper.
Minor errors include:
- Inconsistent terminology: HPV16, HPV-16, HPV type 16,
- Capitalization of plural S (HPV+OSCCS instead of HPV+OSCCs)
- Line 282: “91 to 100%” should be changed to either “91% to 100%” or “91-100%”
- Subject/ verb disagreement
- Lines 283-4: “In addition, four trials phase II trials assessed the efficacy…” Should be changed to, “In addition, four phase II trials assessed the efficacy…”
- Line 303: “HP16” should be changed to “HPV16”
- Non-inferiority is used 3 different ways in the paper: noninferiority, non inferiority, and non-inferiority. I believe the most appropriate is non-inferiority. Whichever is chosen, should be used consistently throughout.
- “On the other hand…” or “On the contrary…” are used 11 times within 6 pages of the text. I would suggest adding additional/alternative transitional phrases.
- Other grammatical errors (spacing, commas, etc.)
Reviewer 2 Report
I was surprised to note that the paper by A.G. Schache et al in Clin Cancer Res 17:1-10 (2011) was not referred to. While I agree this data was based on a smaller cohort than in Augustin et al, it described the advantages and disadvantages of different HPV testing modalities significantly earlier than 2018
The authors should be clear that the link between OPSCC and cervical HPV infection, or indeed sexual behaviour in general, is not proven. For example, see the meta analysis by JA Chancellor et al 2017
Treatment of HPV+ve OSCCs varies by country and it is not true that the ‘gold-standard’ is chemoradiotherapy. The authors should acknowledge national differences in treatment recommendations.
Round 2
Reviewer 1 Report
The authors have addressed all of the issues with the first version of the manuscript.